# Know Your Customer (KYC) Implementation with Smart Contracts on a Privacy-Oriented Decentralized Architecture

**Nikolaos Kapsoulis \*, Alexandros Psychas \*, Georgios Palaiokrassas \*, Achilleas Marinakis, Antonios Litke and Theodora Varvarigou**

Electrical and Computer Engineering school, National Technical University of Athens, 157 80 Athens, Greece; achmarin@mail.ntua.gr (A.M.); litke@mail.ntua.gr (A.L.); dora@telecom.ntua.gr (T.V.)

**\*** Correspondence: nkapsoulis@mail.ntua.gr (N.K.), alps@mail.ntua.gr (A.P.), geopal@mail.ntua.gr (G.P.)

**Abstract:** Enterprise blockchain solutions attempt to solve the crucial matter of user privacy, albeit that blockchain was initially directed towards full transparency. In the context of Know Your Customer (KYC) standardization, a decentralized schema that enables user privacy protection on enterprise blockchains is proposed with two types of developed smart contracts. Through the public KYC smart contract, a user registers and uploads their KYC information to the exploited IPFS storage, actions interpreted in blockchain transactions on the permissioned blockchain of Alastria Network. Furthermore, through the public KYC smart contract, an admin user approves or rejects the validity and expiration date of the initial user's KYC documents. Inside the private KYC smart contract, CRUD (Create, read, update and delete) operations for the KYC file repository occur. The presented system introduces effectiveness and time efficiency of operations through its schema simplicity and smart integration of the different technology modules and components. This developed scheme focuses on blockchain technology as the most important and critical part of the architecture and tends to accomplish an optimal schema clarity.

**Keywords:** permissioned blockchains; know your customer; user privacy protection; enterprise blockchains; business intelligent smart contracts

---

## 1. Introduction

Digital technologies have transformed media content production and distribution in the global entertainment and media industry over the last two decades. Many challenges still remain, though, especially relating to the way in which digital content can be distributed on the Internet, and how content contributors are compensated when their materials are used or bought through legitimate channels. One of the latest experiments in this specific industry involves the use of blockchains and cryptocurrencies for micropayments—payments by consumers of very small sums of money to access specific content. Blockchain technology and cryptocurrencies can provide an ideal, cost-effective framework for payments, preserving privacy, with low commission fees and instant financial transactions without intermediaries.

The blockchain could also store instructions, in the form of a smart contract, for how the content creators would be compensated for the song or music, and how the users can access it. The innovative features that can be offered with blockchain technology are, among others, the ability to have highly personalized services, whereby user IDs, profiles, preferences and history will be present whenever needed by the users and for every media content that is created, consumed, shared, recommended or further analyzed, and the sustainability of media content and services since the blockchain

technology can ensure a long-lasting, scalable and disruptive revenue-based business model that can enhance the financial and technical sustainability of media and services.

The blockchains rely on digital signatures (based on cryptography) to define the identities of the participants in the network. In the Bitcoin network, for instance, the wallet ID is the one which defines the identification of the participant and, through this, someone can search for specific transactions and interact with him/her. Creating a digital ID, combining the decentralized blockchain principle with identity verification, would act as a digital watermark, assigned to every online transaction. This would allow organizations to check identity on every transaction in real time, virtually eliminating fraud. Blockchain's decentralized approach can give control back to users, preventing fraud and boosting trust in the process. However, the blockchain approach on identity management is rather narrow in scope as it does not provide full-flavored identity management functionality and possibilities to interact with third party off-chain services. This is a research topic which is handled in the current paper.

Know Your Customer (KYC) processes strongly rely on identity management, and provide the backbone of organizational and financial institutions' anti-money laundering efforts. Especially, to date, with many applications in financial technology being implemented on blockchains, KYC processes will have to play a significant role in trying to couple valid identity management with privacy-preserving techniques in order to be able to be in compliance with regulations such as General Data Protection Regulation (GDPR). This area of research is the main focus of the paper. In particular, the motivation of the current research is to design and implement new, efficient and effective KYC processes which will run in a decentralized mode over blockchains. For this reason we design an overall architecture which makes use of highly decentralized technologies such as InterPlanetary File System (IPFS) and Quorum blockchain that allows, through the suggested developed smart contracts, the implementation of multi-party KYC processes on top of blockchains. The specific implementation along with the description of the use case through which it has been validated has been part of the Bloomen project which is described briefly in the following.

*Blockchains Transforming the Media Experience: The Example of Bloomen*

The increasing use of digital technologies for media content production, distribution and delivery has enabled the design of new solutions. Blockchain technology can provide a novel framework allowing privacy preservation, copyright protection and new models for direct compensation of certified content creators. An example of such an approach for disrupting the media industry with blockchain technology is the Bloomen project (European Union's Horizon 2020 research and innovation program under grant agreement No. 762091 Bloomen: Blockchains in the new era of participatory media experience. website: http://bloomen.io/). The main goal of the Bloomen project is to extend the use of the blockchain technology to handle different online user transactions, providing an innovative way of content creation, sharing, personalized consumption, monetization and copyrighting. A key part of this approach is the KYC process, in order to link the cryptographic identities to real-world identities and to have access to a permissioned blockchain network.

Identity management issues handled within Bloomen include both identity verification across different networks (outside and inside the blockchain) and identity management over closed silos systems and open public ledgers. Given the decentralized nature of the architecture proposed by the project, it addresses how to distribute and decentralize the identity management functionality, including the options for users and other third parties to set this functionality themselves, as opposed to depending on central systems administrators. Moreover, it addresses how to distribute the identity management functionality across the different levels of the Bloomen architecture (e.g., within the blockchain and the application level) and investigates how, with the complexity induced by the decentralization, identity management can be made to be universal, i.e., how it can extend across very dissimilar use cases. The context of the research work presented in this paper has been implemented in the frame of the Bloomen project as a way to implement a lightweight, efficient and

decentralized KYC process on a Quorum blockchain in order to facilitate the provision of media applications.

The rest of the paper is structured as follows: Section 2 presents a description of work related to blockchain technology, distributed applications and traditional KYC approaches. A detailed analysis of our proposed architecture and implementation details are provided in Section 3. Finally, Section 4 is devoted to discussion and conclusions of our work, while Section 5 presents future extensions of our system.

## 2. Related Work

Blockchains are currently getting a lot of attention as a distributed way of storing data. However, the irreversibility and transparency of blockchains mean they are probably unsuitable for personal data. As far as data privacy is concerned, data stored in blockchains cannot be changed, which means that personal data they contain cannot be removed and so it is very important to design blockchains in order to protect users' privacy. One approach could be to have blockchains used only to provide a timestamp for information held elsewhere. If specific content needs to be taken down from a public source, the fact that the content existed at a given point would still remain in the blockchain but the stored hash would now point to other content that has been changed or simply removed. This approach of using blockchains purely as a time-stamping mechanism and not as a data store has the additional benefit of being more likely to scale in the face of large amounts of data needing to be recorded. Finally, additional encryption of data before it is pushed into the blockchain can be possible. The main problem with this approach is that if the decryption key for encrypted data is ever made public, the encrypted content is readable by anyone with that key; there is no way of encrypting the data with a different key once it is embedded within the blockchain. Regardless of the approach taken to designing blockchains, every blockchain contains transaction data and thus all privacy by design principles have to be taken into consideration before letting any transaction into the ledgers of public/private blockchain implementations.

The underlying technology and ideas behind blockchain have significantly evolved since it was originally proposed as an accounting method for Bitcoin cryptocurrency [1]. Naughton [2] suggests that blockchain technology could be "the most important IT invention of our age", while Mougayar [3] says that it is "at the same level as the World Wide Web in terms of importance". A milestone for the course of blockchain technology was the development of the Ethereum project, offering new solutions by enabling smart contract implementation and execution. It is a suite of tools and protocols for the creation and operation of Decentralized Applications (DApps), "applications that run exactly as programmed without any possibility of downtime, censorship, fraud or third-party interference". It also supports a contract-oriented, high-level, Turing-complete programming language [4], allowing anyone to write smart contracts and create DApps.

In recent years, after realizing that its potential goes beyond cryptocurrencies [5], a lot of related research has been conducted. Blockchain technology and smart contracts as well leverage the underlying technology, having applications in several domains. Smart contracts in Ethereum are (mainly) written in the programming language Solidity [6] and numerous distributed applications (DApps) have been proposed in research works such as for the government sector [7,8], funding mechanisms [9] and many more. Distributed applications have been introduced, successfully blending Blockchain with Internet of Things (IoT) [10]. Decentralized applications have been proposed based on blockchain technology for sharing Internet of Things (IoT) sensors' data [11]. Papadodimas et al. [12] presented a platform for sharing (buying and selling) measurements of IoT weather sensors operating on the Ethereum blockchain, acting as a marketplace for IoT sensor data, exploiting the Sensing-as-a-Service (S2aaS) business model [9], which blends blockchain and IoT [13] for monetization and extracting value from IoT data.

Over the last few years, many digital cryptocurrencies have also been introduced, as also presented on a technical survey on digital cryptocurrencies [5] and smart contracts implementing value tokens as well. In [14], the use of different smart contracts (including a payment token and an asset manager) in the media industry, combining blockchain, web technologies and user-generated

multimedia content, allowing direct monetization for the content creator, is examined. A huge number of financial banking transactions take place every day. It is indicative that in July 2019 the Society for Worldwide Interbank Financial Telecommunication (SWIFT) recorded an average of approximately 32 million transactions per day [15,16]. Blockchain can enable parties with no particular trust in each other to exchange digital data on a peer-to-peer basis with fewer or no third parties or intermediaries. In the recent report Scientific and Technical Research Report of European Commission on Blockchain [17], the need for Know Your Customer mechanisms is highlighted: "the obligation of cryptocurrency exchanges and custodian wallet providers within the scope of EU regulation to implement mechanisms to counter money laundering and terrorist fundraising, such as 'know your customer' (KYC)". Blockchain-based identity management and authentication frameworks have been proposed [18,19] recognizing the potential of decentralizing the ownership of credentials and offering a universally available protocol for verifying one's record in an immutable chain of data. Mikula et al. [20] proposed an authorization and authentication proof-of-concept system for identity management based on a use case concerning Electronic Health Records, where an immutable and auditable history is desired for data concerning patients. Widick et al. [21] presented a blockchain-based authentication and authorization framework to control access to the resources of an IoT device, while Mudliar et al. [22] examine the integration of national identity with blockchain technology.

It is evident that previously mentioned works involve value exchange through blockchain transactions and dedicated created smart contracts, making Know Your Customer processes necessary. In this direction, a system is presented that successfully blends smart contracts for exchanging value in the media industry in a decentralized manner, integrating a KYC process handling on-chain and off-chain data. Recent works have tried to tackle the problem of data management and KYC for blockchain applications. Shabair et al. [23] introduced a blockchain-based KYC proof-of-concept system and an orchestration tool for managing private blockchain environments over large-scale testbeds. In their work they highlight the need for additional research on security and privacy issues of blockchain applications. Norvill et al. [24] presented a demo of a system that allows automation and permissioned document sharing in order to simplify and reduce the work required by the KYC process, while Zhang and Yin [25] conducted research on a digital copyright management system based on blockchain technology. They focused mostly on a PBFT consensus mechanism improved by Tendermint [26], replacing the original Ethereum POW, digital signatures and smart contracts to design user account management strategies, copyright review, and applications for the needs of digital rights management. In this work, the design and implementation of smart contracts for the KYC process on a decentralized approach are further explored.

Blockchain is beginning to transform industries and there is an increasing interest in exploring its potential for various production use cases, especially for supporting multi-party processes where members do not necessarily trust each other. However, there are many challenges that remain to be addressed such as trade-offs between respecting privacy and supporting transparency. Bhsaskaran et al. [27] described the design of smart contracts for consent-driven and double-blind data sharing on the Hyperledger Fabric blockchain platform [28] into a KYC application, where the data are submitted, validated and kept within the ledger, supporting different consent rules and privacy levels. Vishwa et al. [15] presented a decentralized data management system for data privacy and control, focusing on multimedia files. In their solution they use an external data lake, namely a centralized data storage solution on a cloud to store the transaction details of all the data added on the blockchain. In order to access the blockchain, a user signs up by broadcasting their identity and will be accepted by the consent of the majority of the nodes and will be provided a new identity and access permissions. In the presented approach, the additional use of IPFS leads to a decentralized application while there is successful implementation of smart contracts and software components that leverage blockchain to automate tasks related to the KYC process.

## 3. Authorization alongside Know Your Customer

In this section, the concept and execution of KYC standard's user authorization inside a blockchain environment is analyzed. User privacy protection is implemented in a permissioned blockchain method through two types of smart contracts. The public "KYC Smart Contract" is designed to perform CRUD (Create, read, update and delete) operations (blockchain transactions) of KYC-approved users and the private "KYC Admin Smart Contract" submits transactions concerning the CRUD operations for the KYC file repository, i.e., InterPlanetary File System (IPFS). The integration of blockchain with the KYC standards, together with the well-developed business-logic smart contracts, introduces a simple enterprise-focused and time-efficient schema. The section is divided in subsections describing the on-chain Know Your Customer authorization process with regard to user privacy requirements, the application architecture, the smart contract implementations and the use case results.

### 3.1. KYC and Privacy Requirements

Know Your Customer procedures can keep companies' and consortiums' permissioned blockchain networks more secure when registering different clients from all over the globe with different regulations and laws. In order to be safer though, certain conditions must be met with regard to user privacy. User registration constitutes the first step on the customer on-boarding process. When a legitimate client is filling their information on the KYC Registration scheme of the broader blockchain ecosystem, several things happen naturally. The user authorization process to the permissioned blockchain network of a company or a consortium is bound by certain security measures that the KYC mechanism maintains. For instance, the one or union of corporations that own the blockchain network determine that a user is a legal citizen and has a suitable and acceptable background to enter the network. At this point, the external entity is introduced (see "KYC Document Evaluator" in "Section 3.2 Architectural Approach" below). Its role of a third-party non-profit validator that approves the KYC documents submitted adds simplicity to the system. In that sense, the business network is secured from money laundering mechanisms, identity theft incidents and global terrorist finance crime. Such terms may not seem obvious at first; they are comprised of the most important ones though, when it comes to signed users executing blockchain transactions inside global enterprise networks that use cryptocurrencies or not. At the same time, the users themselves do not feel that their personal information is under any danger of being shared or even sold to third-party companies or intelligence divisions. In this structure, their identity information, family or property status or even private financial records that may be requested during authorizing and entering such a system are fully protected against illicit trade and immoral data exchanges.

Moreover, on the decentralized architecture of the system there exists a decentralized way of storing the KYC-related documents of the network members. In this decentralized storage, BitTorrent-like peer-to-peer architecture, customer data privacy is considered of vital significance. Customer information is sealed with a one-way function policy (hash). Eventually, the one that possesses that data and the one-way function can read the whole information while not being able to access it physically, though. Simultaneously, blockchain compatibility is offered with ease.

An important part of the whole system's development constitutes the users' persistence in the network. Customers should remain valid as users inside the blockchain network for a certain predefined period of time. At the end of it, they are no longer valid and they are excluded. Scenarios of excluding the user occur after a certain number of adequate warnings and options of expanding their expiry date with re-examination of newly submitted KYC documents.

### 3.2. Architectural Approach

Using the current situation in the KYC systems as well as the privacy requirements that arose in the process of establishing this type of software (as described in Section 3.1) as a guideline, the blockchain-based KYC component was envisioned and established. In the figure below (Figure 1),

the development of the application architecture and the individual processes elaboration is demonstrated in a vector graphics diagram.

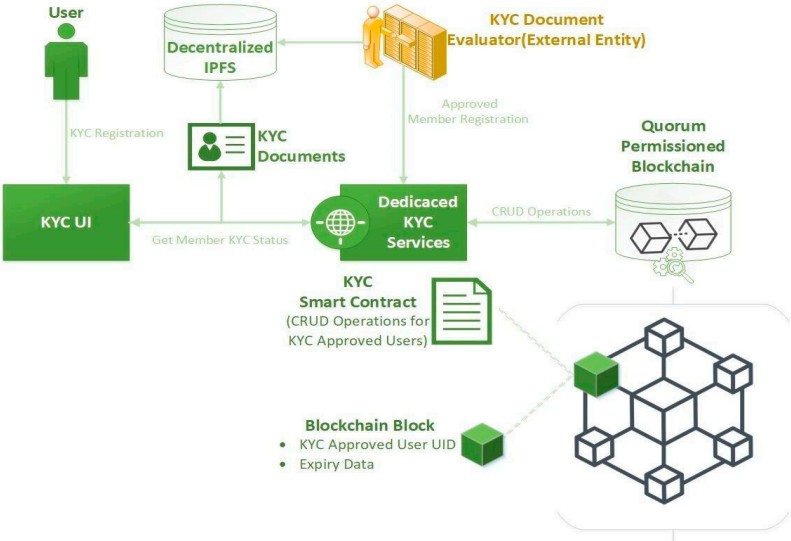

**Figure 1.** Architectural development and processes elaboration.

The whole system's architectural approach focuses on simplicity of operations and positions blockchain technology under the spotlight. Within the logical flow of the procedure, all aspects and functionalities of the modules and the entities that compose the system are explained in detail. Firstly, the user, a candidate system customer, submits their Know Your Customer personal documentation during the "KYC Registration" process through the dedicated and simple user interface called "KYC UI" (Figure 1). Obviously, it is the responsibility of the user to properly describe the demanded information; it is necessary to notice that a new scanning mechanism will be sensitive for misbehaving users and will exclude them early on in the process (see "Dedicated KYC Services" below).

After proper document submission, beyond the focus of this paper, an easy and simple-step procedure, the KYC Documentation is stored in a decentralized peer-to-peer blockchain-compatible repository, IPFS (InterPlanetary File System https://ipfs.io). IPFS protocol creates a resilient system of file storage on a decentralized peer-to-peer network. Though, IPFS has simple security over its data. The core security tools behind IPFS technology consist of cryptographic hashing techniques and mechanisms. Every chunk of data that is stored on IPFS has a dedicated address that originates from a special data-hashing procedure. It is known that a hashing process consists of a one-way function that takes as input the data and outputs a single hash, while there is no reverse function. Similarly, IPFS content addressing associates a single hash (content id or CID, e.g., "QmbFULfor2mTXvsoBPS9EkoC5Xzt56geutPpm49dqpjz8Y") that routes to the content data. In order to access the content data, its CID is needed.

Following the successful storage and validation of the KYC data, a procedure is triggered in order to store specific information to the blockchain. The data that will be stored in the blockchain is only the essential information in order to identify the KYC-approved user, and also the time period in which the KYC approval is valid. Therefore, there is no sensitive information stored in the blockchain. In this way, sensitive information of the user cannot be accessed from the blockchain members, only the validity of the member. The next step in the process is interacting with the blockchain and storing the data just mentioned.

Interacting with the blockchain requires specific libraries and drivers, which connect to it and enable the exchange of information. Dedicated KYC Services is a component that contains all the

appropriate software and drivers in order to connect and interact with the blockchain and by extension with the smart contract.

As far as the blockchain implementation is concerned, a Quorum blockchain was used in the system. One of the main reasons for using the Quorum solution was that it is based on an existing well-established blockchain, Ethereum [4,29]. Furthermore, being able to produce smart contracts using Solidity, and on top of that using the security and permission features of Quorum, were crucial functionalities that led to the selection of Quorum. As it was mentioned in the previous paragraph, the data stored in the blockchain and more specifically inside the smart contract are redundant enough in order to identify the user but ensure anonymity. Communicating with the smart contract requires the development of specific functions that can be accessed by the Dedicated KYC Services and return the data needed to confirm the KYC approval for the specific user account.

### 3.3. Implementation of Smart Contracts with IPFS Storage

Both IPFS- and Quorum-permissioned blockchain were used in order to create a decentralized system that not only preserves but enhances the privacy and security of the user's personal information. As described in the previous section, IPFS is the repository in which all the user information is stored. IPFS database fragments the KYC files and produces an endpoint object in order to link all the parts of the user's KYC files (Figure 2). In order to have access to the data, a person must know the address of this endpoint object. Quorum-permissioned blockchain enables the creation of private/permissioned blocks that contain smart contracts only specific users have access to. In the rest of the blockchain only the hash of these private contracts and blocks is broadcast, in order to validate the integrity of these blocks by the whole blockchain.

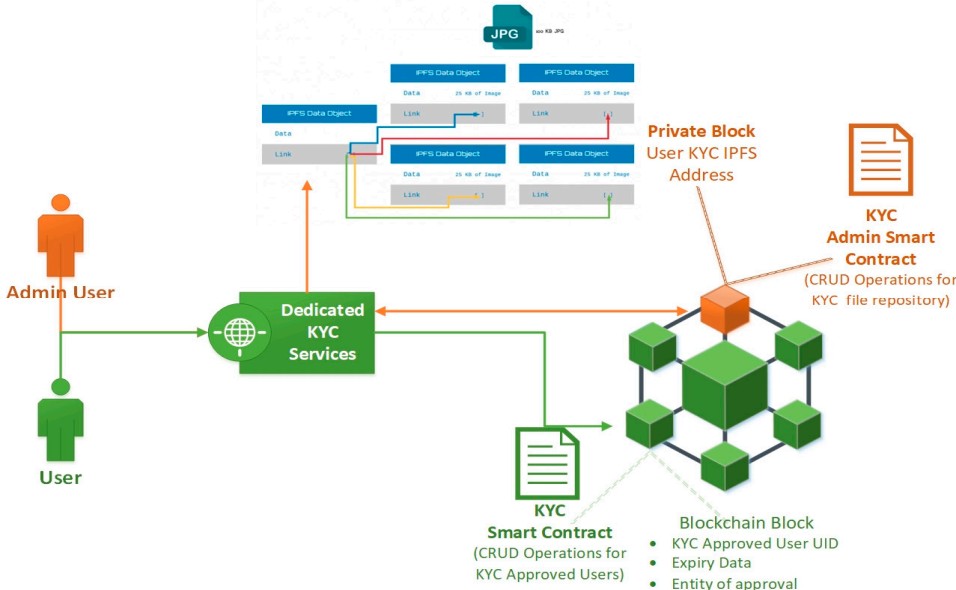

**Figure 2.** Public (user) and private (admin) Know Your Customer (KYC) smart contracts.

As depicted in the image above (Figure 2) the Quorum blockchain contains private and public smart contracts, both giving access to different levels of information. As far as the public KYC contract is concerned, it contains all the functions (Table 1) in order to enable the CRUD operations for the user information stored inside the smart contract 'structs'.

**Table 1.** Public smart contract methods description.

| Function Name | Description | Input | Response |
|---|---|---|---|
| GetKYCMemberApproval() | This function is responsible for delivering basic KYC information upon request. All the data returned from this function do not contain any sensitive information about the user, only the minimum amount of information in order to identify the User and the time period this Member is approved for. | **Address**: User account address | **Date**: Expiration date of the KYC approval.<br><br>**String**: Entity that approved the member |
| UpdateKYCMember() | KYC approval of an account needs to be updated when the approved period has passed, and also there are cases in which an approved member needs to be banned due to malicious activities. This function is responsible for updating the KYC information stored in the blockchain. | **Address**: User account address<br>**Date**: Renewed expiration date<br>**String**: Entity that approved the member<br>**String**: Admin account private key | Success message |
| CreateKYCMember() | When a new member is approved by the system, the KYC info must be stored in a structure inside the Smart Contract. This method is responsible for creating a new record with the KYC info of a new member. | **Address**: User account address<br>**Date**: Renewed expiration date<br>**String**: Entity that approved the member<br>**Bytes32**: Admin account private key | Success message |

Private smart contracts give access to all the KYC documents stored in the IPFS, for evaluation or update purposes. In order to have access to the user's KYC documents in IPFS, the admin users need a specific CID (Content Identifiers). This CID is assigned by the IPFS to the folder that contains all the KYC files of a specific user. IPFS links all these files together, and in this way admin users can access and modify all the KYC files using a single entry point (Figure 3).

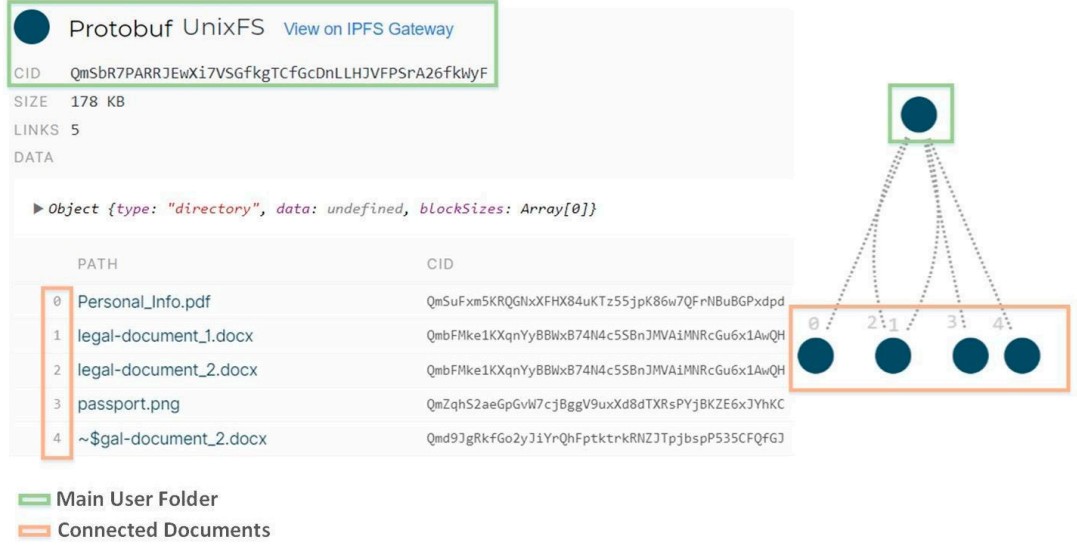

**Figure 3.** IPFS content-addressed file system.

These CIDs are stored safely in the private smart contract and only the admin users, with specific authorization, can call the methods which return the CID of users in order to have access to these files. Functions described in the table below (Table 2) are accessed only by specific nodes in the blockchain with administrator privileges.

**Table 2.** Private smart contract methods description.

| Function Name | Description | Input | Response |
|---|---|---|---|
| GetUserCID() | This function is responsible for returning the CID of the folder containing the KYC documents. It is important to mention that in order to access the functionality of this method, the smart contract requires the user accessing this method to be in the pool of users that have security clearance to access this smart contract. Moreover, the function also requires the hashed private key of the user accessing this method. | **String**: User account ID<br>**String**: Admin hashed private key | **String**: Corresponding CID |
| CreateUserCID() | When the KYC documents are successfully stored in the IPFS, this method is called to store the KYC CID in order to make them discoverable and accessible to the admin users of the private smart contract. | **String**: User account ID<br>**String**: Folder CID<br>**String**: Admin hashed private key | Success message |

*3.4. Use Case Results*

In this section, the use case implementation and results are presented through a gradual description of the system's procedure. In the context of an enterprise's blockchain system where every new user (i.e., an organization as a customer) needs to be thoroughly identified for their

validity, the proposed KYC privacy-oriented architecture implements a simple and secure practice valuable for user data protection management in permissioned blockchain networks. The main issue it tries to address is that permissioned blockchain networks offer no user data privacy among blockchain network participants when it comes to sensitive user data, such as financial records, family status, etc. The proposed architecture provides an implementation and automation of sensitive user data protection thus high-level user privacy inside a permissioned blockchain network. This implementation and automation is achieved through the development of well-structured and encapsulating business intelligence smart contracts. The outcome of the whole system as an operation procedure produces a fully functional scheme that can benefit any consortium that requires data privacy in their blockchain network, while from a technical research point of view, it consolidates various contemporary technologies (e.g., permissioned blockchains, IPFS).

The following progressive queue of processes describes the use case implementation and results. Firstly, a new user provides their information to the "KYC UI" (Figure 1); this data includes i) uploading of their Know Your Customer documents and ii) defining the requested valid period that they will be bound to if accepted (i.e., expiration date until which they are eligible to be part of the blockchain network). Now, the External Entity ("KYC Document Evaluator") examines these documents and if confirmed for their validity, the private smart contract method "CreateUserCID()" is called by an admin account to begin the user-approval process. Afterwards, the KYC docs are stored in the IPFS repository (a decentralized file system) and through the private contract method "GetUserCID()", the corresponding content-identifier (CID) is returned to the admin profile. Now, the user is approved to enter the permissioned blockchain network. In parallel, the public contract method "CreateKYCMember()" creates the new member's network information and sets their expiration date. These are stored on-chain in the corresponding smart contract member struct (storage). Now the new user is granted access to the permissioned network.

When user identification and expiration date check is needed by an admin member, public contract "GetKYCMemberApproval()" is called and only this exact information is returned (i.e., from the IPFS decentralized storage) together with the name of the External Entity that approved the member (no sensitive information of the user such as family status, financial records etc. is included in the reply); this response information is stored in the smart contract storage (Figure 4):

```
struct registeredMember {
    address memberAddr;
    uint256 time;
    string approveEntity;
}
```

**Figure 4.** Member non-sensitive information is stored in the blockchain block (on-chain); sensitive information is stored on the content-addressing IPFS file-system and accessed only by authorized admin members.

When a member's approved period passes or in case they are misbehaving inside the network, the public contract method "UpdateKYCMember()" should be called by an admin account to renew or shorten the validation period. Now, this function either directly updates a member's fields of "time" and "approveEntity", which happens when the expiration date of a member is extended by the same entity that approved them in the first place or a different one, or dismisses misbehaving members by updating their validation period to an old one. These smart contract methods add to each and every process executing inside the proposed system.

In Figure 5, the two smart contracts, the public and the private, are presented by a monitoring application's point of view.

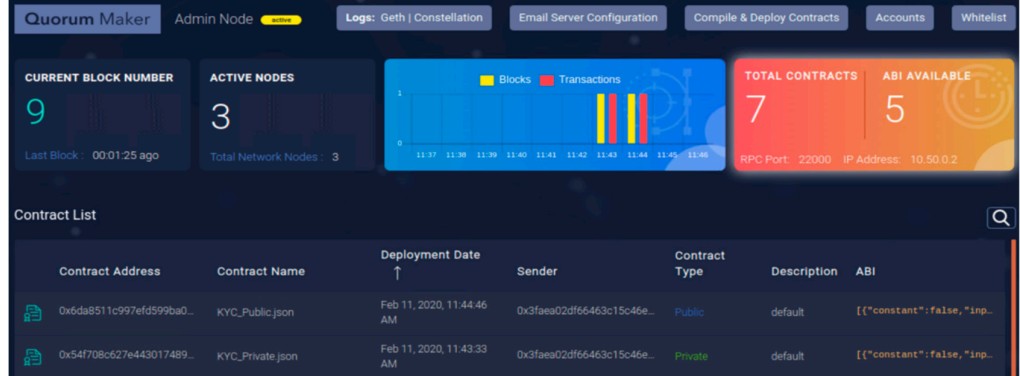

**Figure 5.** The public "KYC Smart Contract" and the private "KYC Admin Smart Contract" deployed on the Alastria Network through the Quorum Maker Utility [30] (to monitor smart contracts).

On the technical side, the developed smart contracts were deployed to a well-known, by the time of writing, public-permissioned blockchain network, the Alastria Network, built on the Ethereum-based blockchain implementation, Quorum, and being a public network with necessary permissions (permissioned-type) to enter and run decentralized applications (DApps). Additionally, "Truffle Suite" is used for the compilation and deployment of the smart contracts while the Solidity version used is ^0.5.11.

The whole project's process of Quorum and IPFS integration presented an interesting amount of research work and it constitutes a valuable future course for storing user information and managing through blockchain.

## 4. Discussion and Conclusions

Privacy and security issues on blockchains are important features for any industry-focused applications and play a very important role especially as they often concern information shared over media channels (such as social media), data that contain personal preferences over media items, or interactions between users. Blockchains are supposed to be used mainly as a distributed way of storing and sharing data, but the irreversibility and transparency of blockchains mean they are probably unsuitable for privacy-sensitive data (data that can reveal physical identities and disclose for instance consumer habits, privacy and proof of location etc.). Data stored in blockchains cannot be changed, and so it is very important that specific blockchains are designed to protect users' privacy. One approach could be to have blockchains used only to provide a timestamp for information of specific parts of workflows referring to data that are held in external data repositories. This approach of using blockchains purely as a timestamping mechanism and not as a data store has the additional benefit of being more likely to scale in the face of large amounts of data needing to be recorded. Finally, additional encryption of data before they are pushed into the blockchain can be possible. The main problem with this approach is that if the decryption key for encrypted data is ever made public, the encrypted content is readable by anyone with that key. Regardless of the approach taken to designing blockchains, every blockchain contains transaction data and thus all privacy by design principles have to be taken into consideration before letting any transaction in the ledgers of public/private blockchain implementations. The KYC processes will play an even more dominant role in the way future transactions will be implemented over blockchains. The specific paper has shown how modular, general purpose and easy to manage architectural frameworks can be implemented on top of permissioned blockchains (e.g., Quorum), but with a flavor of an open public blockchain (e.g., through the flavor of solidity based smart contracts) that can make the implementation of KYC processes a reality suitable for a wide set of decentralized applications.

## 5. Future Work

Regarding future extensions of the system, there is work in progress regarding designing an API to enable more data sources to be integrated within our architecture and support more complex procedures for gathering diverse information about specific entities. Additional care will be given to the automatic connection of verified machines and their capability to provide and retrieve information from our deployed smart contracts on blockchain and our overall KYC system.

Different machine learning approaches and artificial intelligence based tools have been proposed to help in the detection of fraud attempts and assist in the whole KYC procedure by automatically processing a large volume of data, for example for image analysis of an uploaded photo and the assessment of the originality of a document. It is interesting to examine how a component or layer with such tools would interact with a blockchain-based KYC system, aiming to improve the quality of the whole procedure. The KYC process sometimes requires complex procedures such as the examination of many external sources and request of additional documents to verify the identity of a person. Additionally, new regulation requirements emerge and the flexibility of a KYC system to introduce new rules is of vital importance. In this direction, there is conscious planning to extend the current research work on smart contracts covering also such multidisciplinary aspects and complex processes.

**Author Contributions:** The authors collaborated on all sections of the paper and all authors participated in all writing and revising. All authors have read and agreed to the published version of the manuscript.

**Funding:** The research leading to these results has received funding from the European Commission under the H2020 Programme's project Bloomen (grant agreement No. 762091).

**Conflicts of Interest:** The authors declare no conflict of interest.

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
