# Peer review of "Know Your Customer (KYC) Implementation with Smart Contracts on a Privacy-Oriented Decentralized Architecture"

_futureinternet, doi:10.3390/fi12020041_

Round 1

Reviewer 1 Report

-The title could be phrased as: "Know Your Customer (KYC) implementation: Smart contracts and a Privacy-oriented decentralised architecture"

-The Abstract is not informative of the contents/findings of the particular paper but rather a generic introduction in blockchain. Please rewrite using shorter sentences and report on your approach, findings, etc.

-Introduction... there is not a single reference in Introduction which could point on where all this information is taken from. KYC is only mentioned towards the end and not in much detail.

-The sub-section in Introduction with the motivating(?) example does not fit. Why is this example here? what purpose does it serve?

-Finishing the Introduction (and Abstract) I am not sure what this paper is about. What is your motivation? what is your approach? why should the reader continue to read the paper? maybe shorten the Introduction and make it convey more about what is to follow and why it is important. Much of the intro could go to related work.

-Again Related work mentions many interesting facts and concepts but does not have a solid narrative. You should better: (a) explain the concepts, (b) current state, (c) challenges, (d) gap-future directions and (e) where your work is positioned (i.e. the last paragraph of the section)

-You start 3.2 "Here is..." ... here where? (not a way to start a sentence)

-In section 3 there is an architecture presented... still it is not clear.. what is the aim... where is the novelty and contribution... section 3 should start with a good overview of these (Following the gap of section 2... here is the authors' proposal)

-section 3.1 presents what the authors are proposing or what is currently is established?

-is section 3.2 the architecture of what's described in 3.1? not clear!

-please do not us first person ("we present below...")

-section 3.4 not sure how the use case is designed.... is it indicative?

-In Funding it shows that this is Bloomen project! Then section 1 is misleading for not reporting that this work is part of this ""example". 

Reviewer 2 Report

Nowadays there are many applications of blockchains, Authors proposed one more. There is a lack of well done literature review. References list covers only 24 positions, And I would suggest to expend the project presentation and add some words on experiments with the proposed solutions.

Round 2

Reviewer 1 Report

The authors have addressed all my comments in an acceptable manner.